# Efficacy of the Multi-Target Compound E153 in Relieving Pain and Pruritus of Different Origins

**DOI:** 10.3390/ph16101481

**Published:** 2023-10-17

**Authors:** Szczepan Mogilski, Monika Kubacka, Artur Świerczek, Elżbieta Wyska, Katarzyna Szczepańska, Jacek Sapa, Katarzyna Kieć-Kononowicz, Dorota Łażewska

**Affiliations:** 1Department of Pharmacodynamics, Jagiellonian University Medical College, Medyczna 9, 30-688 Kraków, Poland; monika.kubacka@uj.edu.pl (M.K.); jacek.sapa@uj.edu.pl (J.S.); 2Department of Pharmacokinetics and Physical Pharmacy, Jagiellonian University Medical College, Medyczna 9, 30-688 Kraków, Poland; artur.swierczek@uj.edu.pl (A.Ś.); e.wyska@uj.edu.pl (E.W.); 3Department of Technology and Biotechnology of Drugs, Jagiellonian University Medical College, Medyczna 9, 30-688 Kraków, Poland; kasia.szczepanska@uj.edu.pl (K.S.); katarzyna.kieckononowicz@uj.edu.pl (K.K.-K.); dorota.lazewska@uj.edu.pl (D.Ł.); 4Department of Medicinal Chemistry, Maj Institute of Pharmacology, Polish Academy of Sciences, Smętna 12, 31-343 Kraków, Poland

**Keywords:** pain, pruritus, itch, histamine H_3_ receptor, sigma receptors

## Abstract

Itch and pain are closely related but distinct sensations that share largely overlapping mediators and receptors. We hypothesized that the novel, multi-target compound E153 has the potential to attenuate pain and pruritus of different origins. After the evaluation of sigma receptor affinity and pharmacokinetic studies, we tested the compound using different procedures and models of pain and pruritus. Additionally, we used pharmacological tools, such as PRE-084, RAMH, JNJ 5207852, and S1RA, to precisely determine the role of histamine H_3_ and sigma 1 receptors in the analgesic and antipruritic effects of the compound. In vitro studies revealed that the test compound had potent affinity for sigma 1 and sigma 2 receptors, moderate affinity for opioid kappa receptors, and no affinity for delta or μ receptors. Pharmacokinetic studies showed that after intraperitoneal administration, the compound was present at high concentrations in both the peripheral tissues and the central nervous system. The blood–brain barrier-penetrating properties indicate its ability to act centrally at the levels of the brain and spinal cord. Furthermore, the test compound attenuated different types of pain, including acute, inflammatory, and neuropathic. It also showed a broad spectrum of antipruritic activity, attenuating histamine-dependent and histamine-independent itching. Finally, we proved that antagonism of both sigma 1 and histamine H_3_ receptors is involved in the analgesic activity of the compound, while the antipruritic effect to a greater extent depends on sigma 1 antagonism.

## 1. Introduction

Pain and itch are different; however, in many ways, similar phenomena, whose evolutionarily developed role is to warn the body against harmful stimuli. The definition describes pain as a subjective, unpleasant sensory, emotional, and cognitive experience associated with actual or perceived tissue damage. This sensation may originate from any body part such as the skin, visceral organs, muscles, or joints [1]. Itch or pruritus is defined as an unpleasant sensation associated with a desire to scratch. It originates from the surface of the skin or mucus [2]. Both may negatively affect the quality of life when they become dominant and persistent. Moreover, the two may coexist in clinical conditions, which usually complicates therapeutic strategies and increases the number of drugs taken, leading to a higher risk of adverse drug interactions. For example, psoriatic patients are the group, in which systemic treatment of pruritus is challenging because of the number of used drugs [3]. Therefore, there is a need for research on agents with both antinociceptive and antipruritic properties. This approach appears to be challenging because basic neurophysiology shows that simultaneous perception of pain inhibits itch and elimination of pain causes excessive itching by releasing its inhibition by pain [4].

In terms of anatomy and physiology, stimuli that cause pain and itch sensations stimulate similar somatosensory pathways, including C fibers of primary sensory afferents in peripheral tissues, conveying the stimuli to the spinal dorsal horn and finally to the brain. Other types of pain and, to a lesser extent, itch fibers include thin myelinated alpha-delta fibers [5]. However, these two sensations are transduced separately by the neuronal nociceptive and pruriceptive subpopulations. This was proved by the finding that the ablation of neurons with expression of Mas-related G protein-coupled receptor member A3 (MrgprA3) reduced scratching evoked by pruritogens but did not affect pain sensitivity [6]. The cell bodies of pain and itch sensory neurons are in the DRGs (dorsal root ganglions) and trigeminal ganglia. The cell body sends out one pseudounipolar axon that bifurcates and extends toward both peripheral dermatomes, which it innervates and the spinal cord for central transmission [7].

The most prevalent pain fibers are the polymodal class of C fibers, which transmit mechanical, chemical, and thermal stimuli. Pain-sensing C fibers may be activated by a range of receptors, including those for inflammatory mediators, temperature, mechanical pressure, and tissue damage. Agents that can cause pain (algogens) by direct stimulation of pain receptors (nociceptors) or by sensitizing these receptors to other stimuli include capsaicin (via TRPV1), histamine (via H1 and H4 receptors), bradykinin (via B1 and B2 receptors), and prostaglandins (via EP1, EP2, EP3, EP4 receptors) among others [8,9,10,11,12]. The diversity of transduction molecules allows for the C class of fibers to have variety in their signal conductance. Subsequent transmission of painful stimuli involves neurotransmitters such as glutamate, as well as other peptides that include the substance P, somatostatin, and calcitonin gene-related peptide (CGRP). The important fact is that, when the body experiences pain, the immune system is activated and primed to heal the damaged tissue, but it also results in the release of inflammatory mediators such as cytokines, which induce peripheral and central sensitization. These mechanisms of sensitization promote pain sensation, which is observed as hyperalgesia or allodynia [9].

Regarding the cause of pain, it can be divided into four main types: nociceptive, inflammatory, neuropathic, and psychogenic [13]. On the other hand, with regard to its duration, a distinction can be made between acute and chronic pain [2].

As previously mentioned, itching shares some similarities with pain; however, it is a distinct sensation governed by different neural mechanisms. Primary somatosensory neurons located in the dorsal root ganglion (DRG) are involved in transduction of the itching stimulus into action potential. Substances or agents that can induce or trigger the sensation of itching by stimulating specific receptors on sensory nerve fibers that transmit itch signals are pruritogens. Some known pruritogens and their corresponding receptors are histamine (via H1 receptors), serotonin (via 5-HT2A receptors), endothelin-1 (via ET-A receptors), prostaglandins (via EP2 and EP4), mast cell proteases (e.g., trypsin, chymase) (via PAR2 (protease-activated receptor 2), ligands for Mas-related G protein-coupled receptors (Mrgprs) (e.g., chloroquine via MrgprA3, opioids via MrgprX2, cholecystokinin (CCK) via cholecystokinin receptor A (CCKAR), bile acids via TGR5 (G protein-coupled bile acid receptor), and IL-31 (interleukin-31) (via IL-31 receptor A) [14,15,16,17].

Itch sensing primary somatosensory neurons can be classified based on their gene expression patterns. Analysis at the single-cell level has identified three subtypes of nonpeptidergic neurons responsible for itch transduction and the early stage of transmission: NP1, NP2, and NP3. NP1 neurons are distinguished by their expression of MrgprD, NP2 by MrgprA3, and NP3 by their expression of serotonin receptors. Notably, NP2 neurons marked by MrgprA3 have been identified as specific to the itch sensation, as their activation solely elicits behaviors associated with itching. These free nerve endings are implicated as the primary cause of pruritus and they establish synaptic connections with GRP+ interneurons within the spinal cord, which in turn communicate with GRPR+ neurons [4,18].

The significance of the GRP+ spinal pathway in the transmission of itch is evident in studies involving the disruption of GRPR signaling or the elimination of GRPR+ neurons, resulting in the complete elimination of responses to itchy stimuli. Projection neurons eventually transmit information about itch stimuli from the periphery to the higher centers of brain. Within this process, the parabrachial nucleus (PBN), located in the brainstem, serves as a pivotal relay center [19].

Itch processing within the spinal cord involves interaction with other sensory modalities. For instance, GRP+ neurons are exposed to both itch and pain stimuli, but they suppress their responses to intense pain through enkaphalin signaling. In contrast, a painful stimulus activates both itch and pain sensory neurons, with the perception limited to pain due to the inhibitory role of pain over itch facilitated by inhibitory interneurons within the spinal cord. One subgroup of these interneurons, identified by the transcription factor BHLHB5, is responsive to various anti-itch stimuli and likely inhibits itching via the release of dynorphin and glycine [5,20].

Similarly, another cluster of inhibitory interneurons, labeled by the NPY neuropeptide, has been implicated in the inhibition of mechanical itch through gentle touch. Descending neuromodulation from higher brain centers also influences itch processing in the dorsal horn of the spinal cord, which resembles the pain modulation mechanism. Modulation of these somatosensory sensations involves both facilitating and inhibiting neurons. For instance, the raphe magnus (NRM) projects serotoninergic signals to the spinal cord, which can stimulate GRPR+ neurons and enhance itch by co-activating 5HTR1A and GRPR receptors [21].

There are four different types of pruritus: cutaneous, neuropathic, neurogenic, and psychogenic [22].

Although pain and pruritus are distinct experiences, they share remarkably similar mechanisms for transmitting signals, leading to a significant overlap in their pro-inflammatory mediators, neurotransmitters, and neuropeptides. Common pro-inflammatory mediators include histamine, serotonin (5-HT), endothelin-1 (ET-1), and prostaglandins (PGs). Both conditions are also influenced by inflammatory agents, such as TNF-α and nerve growth factor (NGF), which contribute to peripheral sensitization. Although the immune system and glial cells play regulatory roles in both pain and pruritus, the specific cell types responsible for each differ: macrophages and dendritic cells primarily contribute to pain, whereas lymphocytes play a more prominent role in pruritus [22].

We hypothesized that the novel, multi-target compound E153 has the potential to attenuate pain and pruritus of different origins. We based our hypothesis on the assumption that controlling complex processes, which involve numerous mediators, may be efficiently provided by chemical substances influencing these processes in many directions. Interestingly, the molecular targets of E153 are not evident or direct in the search for new drugs with analgesic and antipruritic activities. Thus, evaluation of its activity may reveal how specific combinations of mechanisms affect pain and itch, thereby introducing a new approach to dealing with these symptoms.

1-(5-(4-Phenylphenoxy)pentyl)azepane hydrogen oxalate (E153; Figure 1) is a potent H_3_R antagonist with high affinity for human H_3_R (K_i_ = 34 nM). The antagonist potency of E153 was confirmed in vitro in a cAMP accumulation assay (IC_50_ = 9 nM) and in vivo in (R)-(-)-α-methylhistamine-induced dipsogenia in rats (ED_50_ = 1.75 mg/kg). Moreover, E153 had > 640-fold selectivity over human histamine H_4_ receptor (K_i_ = 22 μM) and weak affinity for human histamine H_1_ receptor (K_i_ = 1.27 μM). Other studies showed also low mutagenicity of E153 in the AMES test (non-mutagenic up to 10 μM), low hepatotoxicity (IC_50_ = 12.8 μM), and weak hERG channel inhibition (IC_50_ = 2 μM) [23]. Furthermore, E153 inhibited butyrylcholinesterase and monoamine oxidase B in the submicromolar range (*eq*BuChE IC_50_ = 0.59 μM; hMAO B IC_50_ = 0.24 μM) [24]. Therefore, we chose E153 as a promising but not obvious compound for further evaluations in animal models of pain and pruritus.

## 2. Results

### 2.1. In Vitro Radioligand Binding Studies

Although some molecular targets for E153 have been established so far, we aimed to test the compound for affinity for a few additional molecular targets that can be involved in its analgesic and antipruritic activity to fully understand its mechanism of action. We chose those targets based on the chemical structure of the compound and the analysis of the literature, suggesting which mechanism may be associated with the pharmacological profile of E153 [25]. First, E153 was tested at the concentration of 1 μM. The results are collected in Table 1. E153 showed a significant affinity for human sigma 1 (95%), human sigma 2 (102%), and human kappa (52%) receptors, while there was no activity for human delta (−5%) and human μ receptor (11%). Further evaluation showed a very high affinity of E153 for the human sigma 1 receptor with IC_50_ of 2.4 nM, which is comparable with the results for the reference sigma 1 antagonist, haloperidol (IC_50_ = 2.2 nM).

### 2.2. Pharmacokinetic Studies

The pharmacokinetics of E153 were best described by a two-compartment pharmacokinetic model (Figure 2). As presented in Figure 3, the concentrations of E153 following both routes of administration were measurable up to 5–8 h and the proposed model captured concentration versus time data very well. The values of the pharmacokinetic parameters are presented in Table 2.

As presented in this Table, E153 was rapidly (k_a_ = 2.32 h^−1^) and almost completely (F = 0.9) absorbed from the peritoneal cavity. The latter suggests that the compound is not extensively metabolized by the liver. The elimination half-life calculated based on the elimination rate constant as ln2/k_10_ was relatively long and it equaled 0.693 h. This value may indicate that E153 is rather slowly eliminated from the body, and in humans even higher values are anticipated, as pharmacokinetic parameters, including elimination half-life, are linearly related to the logarithms of body weight. Similarly, the volume of the central compartment is also relatively large, exceeding the volume of blood and the total body water in mice. The volume of distribution at steady state (V_dss_) calculated as (1 + k_12_/k_21_)∙Vc was even higher and it was 17.49 L/kg. Such a high value indicates that the compound is widely distributed to many organs and tissues.

Analysis of brain concentrations revealed that the compound reached high concentrations in brain tissue 0.5–1 h following both routes of administration (Figure 4). C_max_ values following i.v. and i.p. administration of the dose of 5 mg/kg were similar (5645.79 and 4740.92 ng/g, respectively), while after the dose of 15 mg/kg i.p., E153 reached the peak concentration of 11,682.69 ng/g. Brain concentrations of E153 decreased in parallel with serum concentrations. Estimated t_0_._5λz_ were 0.90, 1.03, and 1.46 h for i.v., 5 mg/kg i.p., and 15 mg/kg i.p. doses, respectively. Brain-to-serum AUC ratios for the doses mentioned above were 15.65, 28.49, and 14.68, which indicates an extensive distribution of the studied compound into mouse brain.

Peak concentrations of E153 in the kidneys and lungs were reached faster (that is, 0.25 h after i.v. dosing) than in brain tissue. Of the tissues studied, E153 attained the highest concentrations in lung tissue, whereas the lowest was in the liver (43.13 vs. 5.32 µg/g) (Figure 5). This may suggest that the compound may accumulate in the lungs, which act as a reservoir for E153, slowly releasing it into the bloodstream.

### 2.3. In Vivo Pharmacological Studies

#### 2.3.1. E153 Attenuates Acute, Inflammatory, and Neuropathic Pain

We aimed to use different experimental pain models to evaluate the analgesic activity of E153 in acute, inflammatory, and neuropathic pain. Acute thermal pain, where tissues were not damaged (often identified with nociception) was modeled using the hot-plate test. In contrast, chemically induced acute pain (including neurogenic mechanisms) was modeled using the capsaicin test and the early phase of the formalin test. The late phase of formalin test and carrageenan-induced inflammation modeled persistent pain and hyperalgesia associated with peripheral injury and aseptic inflammation. Finally, the model of neuropathic pain, which results from dysfunction or lesion of the peripheral or central nervous system, was triggered by a single administration of oxaliplatin (OXPT).

##### Hot Plate Test

To evaluate the influence of E153 on acute pain induced by thermal stimulus we tested it in the hot plate test. The test evaluates thermal pain reflexes due to footpad contact with a heated surface. E153 significantly prolonged the latency time of the pain reaction compared to the saline-treated control (17.93 ± 1.04 s) only at highest tested dose of 30 mg/kg (37.60 ± 6.13 s) [F(3,36) = 5.272, *p* < 0.01] (Figure 6).

##### Formalin Test

The formalin test is a useful screening model for testing clinically relevant molecules. In contrast to the hot plate test, this test is used to examine an animal’s sensitivity to noxious chemical stimuli, not thermal one. Subcutaneous injection of formalin results in a focal injury that stimulates and then damages sensory endings. Two distinct phases of the nociceptive response are associated with immediate activation of nociceptors and sensitization of spinal reflex circuits during phase I and phase II responses, respectively. E153 did not significantly affect the duration of the nociceptive response in the acute phase of the formalin test [F(3,31) = 2.037, ns] (Figure 7a), but it significantly attenuated the paw licking or biting behavior in the late phase at all doses tested (10, 15, and 30 mg/kg) [F(3,32) = 27.90, *p* < 0.0001] (Figure 7b).

##### Capsaicin Test

Compound E153 was not active in phase I of the formalin test but showed activity in hot plate test, showing the potential to inhibit the acute pain dependent of heat but not a chemical stimulus. High temperature induces pain by activation of TRPV1 receptors, which are also activated by capsaicin [26]. We decided to evaluate the influence of E153 on TRPV1-dependent pain using the capsaicin-induced pain. The compound significantly decreased the behavior of paw licking or biting in that test at all doses administrated (10, 15, and 30 mg/kg) [F(3,30) = 53.20, *p* < 0.0001] (Figure 8).

##### Carrageenan-Induced Inflammatory Hyperalgesia and Edema

To evaluate the influence of E153 on the symptoms of acute aseptic inflammation, such as swelling (edema) and increased pain sensitivity, we tested it in the carrageenan-induced inflammation model. Subplantar injection of carrageenan significantly induced edema (F(1.538, 7.688) = 171.0, *p* < 0.0001). The paw volume increased form 0.988 ± 0.05 cm^3^ before carrageenan injection to the values of 1.493 ± 0.07 cm^3^, 2.002 ± 0.02 cm^3^, and 2.208 ± 0.05 cm^3^ 1, 2, and 3 h after injection, which corresponds with increases of 51.1%, 102.4%, and 123.5%, respectively.

The temporal progression of rat paw swelling is depicted in Figure 9a, revealing that the E153 compound had a notably reducing effect on paw swelling (F3,20 = 82.33, *p* < 0.0001). This effect displayed a dose-dependent pattern, with the administration of the highest dose, 30 mg/kg, resulting in a paw volume increase of only 27.1%, 45.4%, and 42.6%. Since swelling is a prominent indicator of inflammation, it can be inferred that the E153 compound possesses anti-inflammatory properties. Furthermore, experiments involving an analgesimeter and the plantar test apparatus demonstrated that carrageenan induced substantial mechanical and thermal hyperalgesia (F(1.754, 8.770) = 23.12, *p* < 0.001). This hyperalgesic response was observed in terms of the pain withdrawal threshold for mechanical stimuli (Figure 10b), which decreased from the value of 144.16 ± 4.72 g (baseline) before carrageenan injection to the value of 126.67 g (87.8% of the baseline) 3 h after the injection. E153 at doses of 15 mg/kg and 30 mg/kg increased the pain threshold to 112.5% and 111.8% of their baseline, confirming its analgesic activity in inflammatory pain. In the vehicle-treated group, the response for thermal stimuli observed as the paw withdrawal latency (Figure 10c) decreased from the value of 12.13 ± 1.15 s (baseline) before carrageenan injection to the values of 5.17 ± 1.09 s (42.62% of the baseline), 3.85 ± 0.42 s (31.74% of the baseline), and 3.03 ± 0.45 s (24.98% of the baseline) 1, 2, and 3 h after the injection, respectively. E153 at the doses of 15 mg/kg and 30 mg/kg significantly (F(3,20) = 25.22, *p* < 0.0001) increased the latency of paw withdrawal at all time points. The administration of 15 mg/kg increased paw withdrawal latency to 100.8%, 122.6%, and 122.6% of the baseline 1, 2, and 3 after induction of inflammation, respectively. The administration of 30 mg/kg increased paw withdrawal latency to 130.3%, 114.8%, and 126.6% of the baseline 1, 2, and 3 after induction of inflammation, respectively. The results (the level of pain reactivity overreaching baseline) show that E153 attenuates inflammatory hyperalgesia and presents analgesic activity.

##### Oxaliplatin(OXPT)-Induced Neuropathic Pain

We investigated the influence of compound E153 on OXPT-induced allodynia for mechanical stimuli using the von Frey method. In all tested groups, the administration of OXPT caused a significant decrease in mean force that caused paw withdrawal response measured 3 h and 7 days after OXPT injection, which corresponds with early and late phase of neuropathic pain. In the group treated with E153 at the dose of 10 mg/kg (Figure 10a), the basic value of 2.91 ± 0.05 (baseline) was decreased to the value of 1.75 ± 0.04 (60.13% of the baseline) and 1.79 ± 0.06 (61.51% of the baseline) in early and late phase, respectively. The single administration of E153 partially but significantly reversed the effect of OXPT in early (71.5% of the baseline) but not in late phase (F(2.685, 24.16) = 74.68, *p* < 0.0001). In the group treated with E153 at the dose of 15 mg/kg (Figure 10b), the initial value of 2.96 ± 0.03 was decreased to the value of 1.79 ± 0.06 (60.5% of the baseline) and 1.70 ± 0.06 (57.4% of the baseline) in the early and the late phase, respectively. The single administration of E153 significantly attenuated the effect of OXPT in both the early (80.0% of the baseline) and the late phase (79.4% of the baseline) (F(2.903, 26.13) = 73.12, *p* < 0.0001). In the group treated with E153 at a dose of 30 mg/kg (Figure 10c), the initial value of 2.97 ± 0.06 was decreased to the value of 1.72 ± 0.06 (57.9% of the baseline) and 1.80 ± 0.03 (60.6% of the baseline) in the early and the late phase, respectively. The administration of E153 almost completely reversed the effect of OXPT in both the early (97.9% of the baseline) and the late phase (87.9% of the baseline) (F(2.568, 23.11) = 95.22, *p* < 0.0001).

#### 2.3.2. E153 Attenuates Pruritus of Different Origins

To evaluate the antipruritic effect of E153, we tested it in different models of acute itch, where we used three pruritogens: histamine, chloroquine (CQ), and SLIGRL. Histamine-induced itch primarily results from activation of peripheral histamine H_1_R, while CQ induces itch by activating the MrgprA3 receptors. In contrast, PAR2 and MrgprC11 receptors are involved in SLIGRL-induced itch [18].

##### Histamine-Induced Pruritus

The injection of histamine into the skin’s dermal layer resulted in episodes of scratching bouts at the quantity of 66.50 ± 6.7. Both E153 and reference compound—pyrilamine—significantly reduced scratching behavior (F(4,36) = 40.59, *p* < 0.0001) (Figure 11).

##### Chloroquine (CQ)-Induced Pruritus

Intradermal injection of chloroquine solution (CQ) resulted in robust scratching behavior manifested as 66.33 ± 8.67 scratch bouts during the 30 min long observation. Single administration of E153 at doses 2.5 mg/kg, 5 mg/kg, 10 mg/kg, and 15 mg/kg significantly decreased the number of scratch bouts (F(5,44) = 19.14, *p* < 0.0001) but pyrilamine at the dose of 10 mg/kg was not effective in this assay (Figure 12).

##### SLIGRL-Induced Pruritus

Intradermal injection of SLIGRL evoked scratching behavior—16.63 ± 2.26 scratch bouts in a 45 min long observation. A single administration of E153 at doses of 7.5 mg/kg and 10 mg/kg significantly diminished the scratch bouts (F(4,35) = 11.42, *p* < 0.0001), while E153 at the dose of 5 mg/kg and pyrilamine at the dose of 10 mg/kg were not effective (Figure 13).

#### 2.3.3. PRE-084 and RAMH Attenuate E153 Analgesic Activity

To evaluate the role of sigma 1 receptors and histamine H_3_Rs in the analgesic effect of E153, we compared the activity of the compound administered alone or in combination with the highly selective sigma 1 receptor agonist PRE-084 [27] or the highly selective H_3_R agonist (R)-(-)-α-methylhistamine (RAMH) [28]. We assumed that the analgesic activity of E153 was dependent on the blockade of both the H3R and sigma 1 receptors. We used agonists of these receptors to inhibit each of these mechanisms separately, confirming the involvement of antagonism toward these particular receptors in the final effects of E153. We tested it in the late phase of the formalin test because E153 was not active in the acute phase of the test.

Moreover, to establish the role of the interaction between H_3_R and sigma 1 receptors in formalin-induced pain, we tested two other compounds administered alone or in combination: the highly selective H_3_R antagonist JNJ 5207852 and the selective sigma 1 antagonist S1RA [25].

Figure 14 shows that a single administration of PRE-084 or RAMH at a dose of 15 mg/kg did not change the duration of the nociceptive response in the late phase of the formalin test (F(2,23) = 0.99, ns). The joint administration of PRE-084 at a dose of 15 mg/kg or RAMH at a dose of 15 mg/kg and E153 antagonized the effects of test compound (F(2,23) = 0.99, *p* < 0.1). This indicates that both histamine H_3_R and sigma 1 are involved in the analgesic properties of E153.

Neither JNJ 5207852 nor S1RA significantly affected the duration of the nociceptive response in the acute phase of the formalin test [F(3,31) = 2.037, ns] (Figure 15a). JNJ 5207852 significantly attenuated paw licking and biting behavior in the late phase at doses of 15 and 30 mg/kg, whereas S1RA showed activity at a dose of 30 mg/kg [F(3,32) = 27.90, *p* < 0.0001] (Figure 15b). The profile of the activity of these compounds resembled that of E153 in the formalin test (lack of activity in the acute phase and significant activity in the late phase).

Joint administration of JNJ 5207852 at the active dose (15 mg/kg) and S1RA at an inactive dose (15 mg/kg) resulted in a significantly stronger decrease in nociceptive response compared to saline-treated animals, but the joint administration of JNJ 5207852 at the inactive dose (10 mg/kg) and S1RA at the active dose (30 mg/kg) did not affect the activity of the compounds [F(6,55) = 11.76, *p* < 0.0001] (Figure 16).

The combination of a selective histamine H_3_R antagonist at an active dose with an inactive dose of a sigma 1 receptor antagonist significantly augmented its analgesic activity [F(2,23) = 11.74, *p* < 0.001].

#### 2.3.4. PRE-084 but Not RAMH Attenuates E153 Antipruritic Activity

To evaluate the role of sigma 1 receptors and histamine H_3_Rs in the antipruritic effect of E153, we compared the activity of the compound administered alone or in combination with PRE-084 or (R)-(-)-α-methylhistamine (RAMH) in histamine-induced pruritus.

Administration of PRE-084 or RAMH at a dose of 15 mg/kg did not alter animal scratching behavior (F(2,21) = 0.0095, ns). The joint administration of PRE-084 at a dose of 15 mg/kg and E153 at the active dose of 5 mg/kg resulted in a significant decrease in E153 activity. RAMH at a dose of 15 mg/kg had no effect on the antipruritic activity of E153 (F(2,21) = 15.81, *p* < 0.0001) (Figure 17). This indicates that sigma 1 receptors, rather than histamine H_3_Rs, are involved in the antipruritic properties of E153.

## 3. Discussion

We evaluated the analgesic and antipruritic activities of multi-target compound E153. First, in vitro studies revealed that the test compound had potent affinity for sigma 1 and sigma 2 receptors, moderate affinity for opioid kappa receptors, and no affinity for delta and μ receptors. Pharmacokinetic studies showed that after intraperitoneal administration, the compound was present at high concentrations in both the peripheral tissues and the central nervous system. The concentration–time relationship indicated that the concentration of the test compound was sufficiently high in the time intervals in which the behavioral tests were performed. Moreover, the blood–brain-barrier-penetrating properties indicated its ability to act centrally at the level of the brain and spinal cord. We showed that E153 was effective in decreasing both pain and pruritus. Furthermore, the test compound attenuated different types of pain, including acute, inflammatory, and neuropathic pain. It also showed a broad spectrum of antipruritic activity, attenuating histamine-dependent and histamine-independent itching. We also proved that the analgesic and antipruritic effects do not arise from the potent sedative properties of the compound or from its undesirable influence on motor function (see Appendix A.). Finally, we proved that antagonism of both sigma 1 and histamine H_3_Rs are involved in the analgesic activity of the compound, while the antipruritic effect to a greater extent depends on sigma 1 antagonism. Using highly selective antagonists for sigma 1 and histamine H_3_Rs, we showed that simultaneous blocking of sigma 1 and histamine H_3_Rs leads to more efficient analgesia.

Considering all the in vitro studies carried out for E 153 until now, several basic molecular targets for this compound have been identified, which are responsible for its pharmacological activity. These targets include histamine H_3_Rs, sigma 1 receptors, BuChE, MAO B, and opioid kappa receptors. Since E153 has the highest affinity for H_3_Rs and sigma 1 receptors, we focused on the analysis of these two mechanisms. However, the rest of the targets should not be excluded, as they may be, at least partly, involved in the analgesic and antipruritic effects of the compound.

E153 is a potent antagonist of histamine H_3_Rs antagonist [24]. These receptors play crucial role in the regulation of histamine activity in the organism [29]. Histamine is a biogenic amine that plays significant roles in various physiological processes, including its well-known involvement in allergic and inflammatory responses [30]. All the effects exerted by histamine result from the activation of four types of receptors named H_1_, H_2_, H_3_ and H_4_, which belong to the family of membrane GPCRs [31]. Although histamine is more commonly associated with pruritus, it also has implications in pain perception, both in peripheral tissues and in the central nervous system (CNS) [32,33,34].

Histamine is a canonical pruritogen responsible for histamine-dependent pruritus. It induces itching by activating histamine H_1_ and H_4_ receptors expressed on the skin and mucosa nerve endings of both NP1 and NP3 subtypes of pruritogenic neurons [4]. Moreover, histamine sensitizes these nerve endings, making them more responsive to other itch-inducing stimuli. Separated or simultaneous antagonism of H_1_R and H_4_R effectively block histaminergic itch [35,36,37]. The role of histamine in pain is not as evident as in pruritus. In peripheral tissues, histamine can contribute to pain perception through its pro-inflammatory actions. When tissues are injured or inflamed, mast cells release histamine. Histamine can sensitize nociceptors directly or indirectly through various mechanisms, leading to inflammatory hyperalgesia. This effect of histamine results from stimulation of peripheral H_1_ receptors and subsequent sensitization of transient receptor potential vanilloid 1 (TRPV1) through the PLC/PKC pathways. TRPV1 is expressed in the sensory neurons and functions as a molecular integrator of pain perception. Under inflammatory conditions, many mediators are released and augment TRPV1 function [38,39]. Thus, in peripheral tissues, histamine acts as a pruritogen and algogen, and its final effect depends on the exact site of histamine release. In the superficial parts of the skin and mucosa, it induces pruritus, but in deeper tissues, it induces pain [5].

In the CNS, cell bodies of histaminergic neurons are localized in the tuberomamillary nucleus (TMN) of the posterior hypothalamus. They send projections to almost all areas of the CNS, contributing to the modulation of alertness, sleep/wakefulness, feeding, endocrine, memory, and other processes including pain perception [29,30,32]. Centrally acting histamine is generally considered to have an analgesic effect [33]. Histaminergic neurons in the TNM send descending projections to various parts of the spinal cord and brain regions involved in pain processing. These projections are believed to contribute to inhibition of pain signals in the spinal cord. For example, histamine release in the locus coeruleus (LC) facilitates noradrenaline neurotransmission in the descending antinociceptive pathways, exerting analgesia [40,41].

H_3_Rs predominantly function as presynaptic autoreceptors on histaminergic neurons. These receptors are essential for maintaining a feedback loop that regulates histamine levels [42]. Additionally, H_3_ heteroreceptors located on nonhistaminergic neurons play a negative regulatory role in the release of various neurotransmitters, including acetylcholine, dopamine, serotonin (5—HT), and noradrenaline [32,43,44]. The receptor expression in both the central (CNS) and peripheral (PNS) nervous systems, particularly along the ascending nociceptive pathway and descending pain-control pathway, suggests the strong involvement of them in pain processing. Within the CNS, this receptor has been found in various brain areas, such as thalamus, hypothalamus, prefrontal cortex, and periaqueductal grey area, and in the spinal cord. In the periphery, the expression of H_3_Rs has been identified in DRG, superior cervical ganglia, and dermal tissues [31,32,43]. Despite some concerns, most data reveal that the H_3_R antagonists/inverse agonists rather than agonist have meaningful analgesic activity [40,45,46]. The mechanism of analgesia induced by H_3_R antagonists/inverse agonists results from blocking of the autoinhibitory H_3_R on histaminergic terminals in the pontine locus coeruleus (LC), which receives efferent projections from the TMN. Facilitation of the endogenous release of histamine leads to a subsequent increase in noradrenaline neurotransmission in the descending antinociceptive pathways. Moreover, H_3_R antagonists/inverse agonists induce α2 adrenoceptor desensitization in the LC and spinal cord, which increases noradrenaline release in the terminal area (e.g., the prefrontal cortex or spinal cord) [40,41,47].

Several previous studies have revealed that H_3_R antagonists/inverse agonists such as E-162, GSK189254, GSK334429, S38093, and A—960656 [40,48,49,50] show significant analgesic activity in different pain models. Our studies proved that E153 also had a wide range of analgesic properties and decreased nociceptive responses, especially in models of inflammatory and neuropathic pain of different origins. The mechanisms involved in pain development in all the tests were different. The hot plate test evaluates thermal pain reflexes due to footpad contact with a heated surface and engages mainly physiological mechanisms involved in pain perception. The hot plate procedure enables a complex estimation of nociceptive reactivity because it integrates supraspinal pathways and mechanisms rather than a simple spinal reflex [51]. Subcutaneous injection of formalin results in a focal injury that stimulates and then damages sensory endings. Two distinct phases of nociceptive response are associated with immediate activation of nociceptors and sensitization of spinal reflex circuits during phase I and phase II responses, respectively. The second phase is a response to tissue damage caused by inflammation. Moreover, it has been recently suggested that formalin injection results in pathological changes that resemble those observed in nerve injury and neuropathic pain [52]. Effects caused by the administration of capsaicin resemble the early stage of the formalin stage but depend on TRPV1 rather than TRPA1 activation [26]. In the oxaliplatin-induced model of neuropathic pain, neurons are directly affected by OXPT, which induces central and peripheral sensitization as a consequence of upregulation of TRPM8 and TRPA1 and downregulation of Kv4.3 neuronal channels. As a direct effect of OXPT on voltage-gated Na+ channels, formation of adducts with mitochondrial DNA leads to changes in expression of membrane-associated proteins, such as channels, and to alterations in cellular metabolism, which results in formation of oxygen species among others [53]. In the carrageenan-induced model of inflammation, contrary to, e.g., the formalin test, nociceptive reactions are not related to the direct stimulation of nociceptors but result from the secondary release of the inflammatory mediators such as prostanoids or cytokines from immunocompetent cells [54]. E153 showed versatile analgesic properties, decreased inflammatory hyperalgesia, and neuropathic allodynia. Moreover, the test compound showed anti-inflammatory activity, which could be observed as decreased inflammatory edema in carrageenan-induced inflammation. The compound administered at lower doses did not significantly affect physiological nociceptive mechanisms (hot plate test or the early phase of the formalin test), which should be considered as a beneficial effect because it proves that the compound does not abolish the natural mechanisms of avoiding noxious stimuli while alleviating pathological pain.

Considering the above-mentioned data, we concluded that the analgesic properties of E153 are, at least in part, associated with its antagonism for H_3_R. To finally confirm that hypothesis, we used probably the most versatile and complex method, the formalin test, in which we co-administered E153 and histamine H_3_R selective agonist RAMH. This resulted in a decreased analgesic effect of E153. This clearly shows that the blockade of H_3_R is a significant mechanism for the activity of the compound. Moreover, JNJ 5207852, a highly selective H_3_R antagonist, showed an E153-like profile of analgesic activity in the formalin test, lack of activity in the early phase, and significant activity in the late phase of the test.

The antagonism/inverse agonism of histamine H_3_Rs leads to increased levels of acetylcholine in the CNS [44,55]. This effect, together with the BuChE inhibitory properties of E153, may be responsible, at least in part, for the analgesic activity of the compound. The activation of cholinergic receptors, such as nicotinic (N) and muscarinic (M) receptors, at the level of the CNS modulates the perception of pain in experimental animals by affecting descending inhibitory pain pathways [56]. However, on the current stage of investigation into the pharmacological activity of E153, it remains a hypothesis, which will be tested in the future.

The role of H_3_Rs in pruritus is multifaceted and is not fully understood. Most of the available data suggest that blocking of H_3_Rs promotes itching. It has been shown that intradermal injection of H_3_R antagonists induces scratching behavior, which is dependent on substance P release [57]. Moreover, Rossbach et al. demonstrated that calcium concentration increases in pruriceptive neurons via activation of H_1_R and H_4_R, as well as inhibition of H_3_R. The authors assumed that the decreased threshold in response to H_3_R antagonism activated H_1_R and H_4_R on sensory neurons, which in turn resulted in the excitation of histamine-sensitive afferents and therefore elicited the sensation of itch [36]. All the papers discussed the role of peripherally expressed H_3_R. There are no valuable data evaluating the influence of centrally acting H_3_R antagonists on scratching behavior. We assumed that some similarities between modulatory mechanisms in pain and pruritus transmission in the spinal cord and brain may result in the antipruritic effect of E153. Pharmacokinetic studies have shown that this compound can easily penetrate the blood–brain barrier. E153 efficiently attenuated histamine-dependent and histamine-independent itching behaviors in mice. Comparing the analgesic and antipruritic effects of E153, it is worth emphasizing that the compound showed better efficacy (effect) and potency (lower doses needed to obtain the same result) in decreasing scratching behavior than in decreasing nociceptive behavior. To evaluate the role of H_3_R, we combined E153 and RAMH in a model of histamine-induced pruritus. In contrast to pain, RAMH did not significantly affect the antipruritic effect of E153. Therefore, we concluded that E153 antagonism of H_3_R is not strongly associated with its antipruritic activity, and some other molecular targets for this compound may be responsible for this property. The limitation of our study is that it has only been performed with E153, and the role of central H_3_Rs in itching requires further research using other substances and more advanced methods.

Most behavioral tests indicate that H_3_R antagonists/inverse agonists do not have any antinociceptive effects in naïve animals, suggesting that H_3_R are not involved or tonically activated in nociception, but are critical for pathological pain states associated with tissue damage and inflammation [40,58]. In the hot plate test, where animals’ tissues are intact and pain has a strong nociceptive character, E153 was active only at the highest applied dose. In tests where the chemical substance affected tissues inducing inflammation, the compound was active at a much wider range of doses. Considering the above characteristics of the analgesic activity of the H_3_R antagonist/inverse agonist, it is suggested that E153 has some additional molecular mechanism responsible for its analgesic properties. Riddy et al. showed the high possibility of coexisting histamine H_3_R and sigma 1 affinity in a single compound, which encouraged us to test E153 towards sigma receptor affinity [25].

The investigated compound showed high affinity for sigma 1 receptors (K_i_ = 1.2 nM) and sigma 2 receptors. Relatively little is known about sigma 2 receptors, which have recently been identified as transmembrane proteins expressed in the DRG, including neurons that are involved in pain. It was found that putative agonists of these receptors (e.g., DKR-1005, DKR-1051, UKH-1114) significantly decreased neuropathic mechanical hypersensitivity [59]. Although the involvement of sigma 2 receptors in the analgesic effect of E153 cannot be excluded, we focused on the sigma 1 receptors, since their role in pain perception is quite well established [27,60,61,62].

At the cellular level, sigma 1 receptors are notably concentrated within the endoplasmic reticulum membranes in conjunction with the mitochondria. When cells undergo activation, leading to elevated levels of intracellular Ca^2+^, sigma 1 receptors become active and relocate to different parts of the cell, particularly in proximity to the plasma membrane. In these regions, they physically interact with various membrane proteins [61]. Among these protein associates of sigma 1 receptors are several molecular targets that play a crucial role in the mechanisms of nociception and pain perception, such as ion channels, including NMDA (N-methyl-D-aspartate) subtype glutamate receptors, calcium channels, potassium channels, G protein-coupled receptors such as opioid receptors, and transient receptor potential (TRP) such as TRPV1, TRPA1, and TRPM8 [63]. After their interaction with protein partners, sigma 1 receptors function as regulatory subunits and exert a significant influence on neurotransmission. Moreover, sigma 1 receptors play a key role in the communication between neurons and non-neuronal cells, such as microglia and astrocytes, and they affect the process of peripheral and central neuroinflammation. Thus, these receptors play a significant role in painful pathological neurotransmissions [27,61,64]. Blockade of sigma 1 receptors by sigma 1 antagonists induces ameliorative effects on pain in animal models of inflammation, central and peripheral neuropathy, osteoarthritis, and cancer [27,60,62,64,65].

We showed that the analgesic effect of E153 was also dependent on sigma 1 receptor blockade. Animal pretreatment with sigma 1 agonist PRE-084 [66] significantly attenuated the analgesic effect of E153 in the late phase of the formalin test. Thus, we concluded sigma 1 antagonism was another mechanism, in addition to H_3_R antagonism, involved in the analgesic effect of E153. Moreover, we showed that the combined administration of a highly selective H_3_R antagonist (JNJ 5207852) and a selective sigma 1 antagonist (S1RA) resulted in a greater analgesic effect than when these substances were administered alone (the ineffective dose of S1RA significantly augmented the activity of the efficient dose of JNJ 5207852; see Figure 16). Such synergism of the action of the two separate mechanisms can be extremely beneficial in the context of obtaining an efficient method of pain management. Research on this synergism seems to be a very interesting direction for the future.

The role of sigma 1 receptor affinity in the molecular profile of E153 was also evident in its antipruritic activity. PRE-084 significantly attenuated (but not completely abolished) the anti-itching activity of E153. There is very little information on the influence of sigma 1 receptors on the mechanism of pruritus. We hypothesized that the TRPV1—sigma 1 receptor relationship may be responsible for the antipruritic effect of sigma 1 antagonism. In fact, all itch neurons described in the Introduction express TRPV1. Consistently, mutation of the TRP channels causes pathological itch [18]. TRPV1 plasma membrane levels can be regulated by its interaction with intracellular proteins, including chaperones. TRPV1- dependent pain was decreased through sigma 1 antagonism because of the influence on the physical interaction between TRPV1 and sigma 1 [67]. A similar mechanism may be observed in itch, but this hypothesis needs to be evaluated.

The main limitation of our study is that E153 is a multidirectional compound, including H_3_R, sigma 1 receptors, sigma 2 receptors, opioid kappa receptors, and inhibition of MAO-B and BuChE. We focused on sigma 1- and H_3_R-dependent mechanisms, but additional molecular targets may also be involved in the final effects of the compound [3,4,8,9]. Moreover, we did not test other routes of administration such as oral administration. Changes in the route of administration may affect both the pharmacological profile of the compound and its bioavailability.

In summary, this study provides evidence that E153 exerted both analgesic and antipruritic activities in animal models of pain and pruritus of different origins and mechanisms. The analgesic effect was dependent on H_3_R and sigma 1 receptor blockade, while the inhibition of the pruriceptive response was, at least in part, dependent on sigma 1 blockade.

## 4. Materials and Methods

### 4.1. Drugs

The tested compound E153 was synthesized and pharmacologically profiled in the Department of Technology and Biotechnology of Drugs (Jagiellonian University Medical College, Kraków, Poland) as previously described [25]. PRE-084 (2-[4-morpholinethyl]1-phenylcyclohexanecarboxylate hydrochloride), a selective sigma 1 agonist, and JNJ 5207852 (1-[3-[4-(1-piperidinylmethyl)phenoxy]propyl]piperidine hydrochloride), a selective H3R antagonist, were purchased from Tocris (Cookson Ltd., Bristol, UK). S1RA(4-[2-[[5-methyl-1-(2-naphthalenyl)1H-pyraol-3-yl]oxy]ethyl]-morpholine hydro-chloride), a known selective sigma 1 receptor antagonist was purchased from DC Chemicals (Shanghai, China). H3R agonist (R)-α-methylhistamine (RAMH) was purchased from Sigma-Aldrich (Darmstadt, Germany).

### 4.2. Animals

The experiments were carried out on adult male Albino Swiss mice (CD-1, 18–25 g) and male Wistar rats (Krf:(WI) WU), 180–250 g). The animals were housed in plastic cages in a room at a constant temperature of 20 ± 2 °C, under a light/dark (12:12) cycle and had free access to standard pellet diet and water. The experimental groups consisted of 6–12 animals (experiments in rats *n* = 6; experiments in mice *n* = 8–12), all the animals were used only once, and they were killed by cervical dislocation immediately after the assay. The rats and mice were previously anesthetized with sodium pentobarbital (60 mg/kg and 37 mg/kg, respectively). The minimum number of animals was used needed to obtain definite and normally distributed results with the utilized test. Behavioral measures (such as jumping, hind paw licking or biting, hind paw elevation, or scratching the injected skin, depending on the method) were scored by trained observers. The observers were blinded to experimental conditions, which means they were not aware of the animal group they observed, for example, control or injected with the test or reference compound. The treatment of laboratory animals in the present study was in full accordance with the respective Polish regulations. All procedures were conducted according to guidelines of the ICLAS (International Council on Laboratory Animal Science) and approved by the Local Ethics Committee of the Jagiellonian University in Cracow (105/2016; 279/2019; 139/2017; 179/2017; 546/2021).

### 4.3. In Vitro Radioligand Binding Studies

Compound E153 was tested in commercially available radioligand binding assays conducted in Eurofins Cerep SA (Celle l’Evescault, France).

### 4.4. Pharmacokinetic Studies

#### 4.4.1. Study Design

E-153 was dissolved in 1% Tween 80 for intraperitoneal (i.p.) administration and in a mixture of DMSO:water (1:10, *v*/*v*) for intravenous (i.v.) administration and used within 1 day of preparation. The compound was administered at a dose of 5 mg/kg i.v. and at two doses (5 and 15 mg/kg) i.p. Before the dose (time 0) and up to 8 h at different time points following compound administration, three to five mice per time point were exsanguinated while under light ketamine/xylazine anesthesia. Blood and brains were harvested to the last measured point, whereas liver, kidneys, and lungs were collected at 0.25, 0.5, and 1 h. Serum and tissues were stored at −80 °C until assayed.

#### 4.4.2. Analytical Method

The compound E153 was synthesized in the Department of Technology and Biotechnology of Drugs, Jagiellonian University Medical College, Cracow, Poland. Acetonitrile, ethyl acetate, hexane, methanol, potassium dihydrogen phosphate, and imipramine hydrochloride (internal standard) were purchased from Sigma-Aldrich (Darmstadt, Germany). Sulfuric acid (96%) was from Merck (Darmstadt, Germany). Sodium hydroxide was obtained from J. T. Baker (The Netherlands). Ultrapure deionized water was obtained using a Hydrolab water purification system (Poland). Blank murine blood and tissues, such as liver, kidneys, brain, and lungs were collected from healthy CD-1 mice. Serum was obtained using centrifugation (EBA 12 R, Hettich, Kirchlengern, Germany) of coagulated blood at 2000 x g for 10 min. Serum and tissues were stored at −80 °C until assayed.

The stock solutions of E153 compound (5 mg/10 mL) and IS—imipramine (5 mg/10 mL)—were prepared in methanol and stored at 4 °C (Polar, Warszawa, Poland). The stock solution of E-153 was subsequently diluted with methanol to prepare working solutions in the range of 0.05–500 µg/mL for plasma and tissue calibration samples.

Tissues (liver, kidney, brain, and lung) were homogenized with water in a 1:5 ratio (*w*/*v*). Prior to the extraction, 100 μL of the plasma samples or 500 μL of tissue homogenates containing calibrators or authentic samples were spiked with 10 μL of IS working solution in a snap-cap microcentrifuge tubes and vortexed (Reax top, Heidolph, Schwabach, Germany) for 15 s. The serum and tissue samples were then alkalized with, respectively, 20 μL or 50 μL of 4 M sodium hydroxide solution, mixed briefly on the vortex mixer, and extracted with 1 mL of ethyl acetate/hexane (30:70, *v*/*v*) mixture by mixing for 20 min on a shaker (VXR Vibrax, IKA, Staufen, Germany). Then samples were centrifuged (EBA 12 R, Hettich, Germany) at 8000 g for 15 min and the organic layer of each sample was transferred to a new snap-cap microcentrifuge tube containing 200 μL of a methanol/0.1 M sulfuric acid mixture (1:9, *v*/*v*). Subsequently the samples were vortexed (20 min) and centrifuged again (8000 g, 15 min). Finally, the organic layer was discarded and 10–50 μL of the acidic layer was transferred to the autosampler vials (4 °C).

#### 4.4.3. Chromatographic Conditions

The concentrations of compound E153 in serum and tissues were measured using a HPLC method with fluorescence detection. The HPLC system (LaChrom Elite, Merck Hitachi) consisted of the following components: an L-2130 pump, an L-2200 autosampler with a 100 μL sample loop, an L-2350 oven, and an L-2485 fluorescence detector. Excitation and emission wavelengths were set to 260 nm and 330 nm, respectively. Analysis was performed at a temperature of 25 °C on a 250 mm × 4 mm LiChrospher 100 RP-18 column from Merck Millipore (Germany), filled with 5 μm particles, protected with LiChroCART precolumn (10 μm particle size). The mobile phase containing acetonitrile and 20 mM potassium dihydrogen phosphate buffer (pH 4.5) in a 7:3 (*v*/*v*) ratio was filtered through the 0.45 µm membrane filter (Supelco Inc., Mainz, Germany) before use and pumped at a flow rate of 1.5 mL/min. Under these conditions, the times of retention for IS and compound E-153 were approximately 4.39 and 8.30 min, respectively. No chromatographic interferences were observed. The method was validated for serum, brain, liver kidney, and lungs. Calibration curves were linear in the range from 5 ng/mL to 20 μg/mL for serum and from 100 ng/g to 50 μg/g for tissues. The sample accuracy deviated by less than 15% from the nominal values, except for LLOQ where it was less than 20%. The precision had a relative standard deviation of up to 15%.

#### 4.4.4. Pharmacokinetic Analysis

E153 serum concentration versus time profiles for both i.v. and i.p. routes of administration and all dose levels were simultaneously fitted to obtain a single set of parameters. Both one- and two-compartment pharmacokinetic models were tested. Bioavailability was estimated as a model parameter (F). Brain concentrations of E153 were assessed using non-compartmental analysis. The maximum concentration (C_max_) and the time to reach peak concentration (t_max_) were obtained directly from the concentration–time data. The terminal elimination rate constant (λz) was assessed using linear regression. Terminal half-life (t_0_._5λz)_ was calculated as ln2/λ_z_. The area under the concentration–time curve from 0 to last measured point (AUC_0–last_) was calculated using the linear trapezoidal rule. Non-compartmental and model-dependent analyses were performed using Phoenix WinNonlin version 6.3 (Pharsight Corp., Mountain View, CA, USA). The final pharmacokinetic model was selected on the basis of visual inspection of the fitting, examination of residuals, parameter precision, Akaike Information Criteria, and Schwarz Criteria.

### 4.5. In Vivo Pharmacological Studies

#### 4.5.1. Hot Plate Test

The hot plate test was performed according to classical method described in Eddy and Leimbach with modifications [68]. At the beginning, animals were tested for their baseline pain sensitivity threshold. For further studies, only mice that fulfilled the criterium of pain response ≤30 s were used. After 24 h, the mice were intraperitoneally (i.p.) injected with either E153 or saline, and after 30 min, they were placed in the hot plate apparatus with a heated surface up to 55 °C (Bioseb, Vitrolles, France). The latency to pain response (licking the hind paw or jumping) was measured. The cut-off time was set to up to 60 s to avoid tissue damage, and mice not responding within 60 s were removed from the apparatus and assigned the score of 60 s.

#### 4.5.2. Formalin Test

The mice received an intraperitoneal (i.p.) injection of either the test compound or a control solution. After 30 min, a 20 μL injection of a 2.5% formalin solution was administered into the right hind paw. Right after the formalin injection, the animals were individually placed into glass containers and observed for the following 30 min. During specific time intervals (0–5 min, referred to as Phase I, and 15–30 min, referred to as Phase II, post-formalin injection), the time spent by the animals licking or biting the injected paw was recorded. [52,69].

#### 4.5.3. Capsaicin Test

The mice received an intraperitoneal (i.p.) injection of either the test compound or a control solution. After 30 min, an intraplantar injection of 1.6 µg capsaicin in a 20 μL solution of 0.9% NaCl was administered to the right hind paw. Right after the capsaicin injection, the animals were individually placed into glass containers and observed for the subsequent 5 min. During this observation period, the time that each animal spent licking or biting the injected paw was recorded [69].

#### 4.5.4. Carrageenan-Induced Inflammatory Pain and Edema

The acute, local inflammation and paw edema was induced by subplantar injection of 0.1 mL of 1% carrageenan (made in PBS) into the rat right hind paw. The paw volume was measured using the dislocation of the water column of the plethysmometer (Plethysmometr 7140, Ugo Basile) before and at 1, 2, and 3 h after subplantar injection of carrageenan.

Furthermore, to assess the heightened sensitivity to mechanical stimuli, we employed the pain pressure threshold test. This involved applying increasing pressure to the dorsal surface of the right hind paw using an automated gauge (Analgesy Meter 37215, Ugo Basile, Gemonio, Italy), following the method outlined by Randall and Selitto. The force’s intensity, measured in grams, was recorded when the paw was withdrawn (withdrawal threshold). For each animal, withdrawal thresholds were recorded both prior to the administration of carrageenan and three hours afterward [21]. The results were expressed as a percentage of the initial reaction, with the nociceptive response before carrageenan administration considered as the baseline (100% response).

To evaluate the heightened sensitivity to thermal stimuli, we employed a plantar test apparatus (Commat Ltd., Ankara, Turkey). Rats were individually placed in plexiglass chambers and allowed to acclimatize for 20 to 30 min before testing. A radiant heat source was positioned beneath the chamber floor, directly under the hind paw. The time it took for the paw to withdraw in response to the heat was automatically recorded by a photocell and an electronic timer. The intensity of the radiant heat was adjusted to achieve baseline withdrawal times of 10–15 s, and a preset cut-off time of 30 s was established to prevent tissue damage. Animals that did not respond within 30 s (the cut-off time) were excluded from the analysis. Three successive applications of the heating stimulus were performed, with 1- to 2-minute intervals between them, and the mean of these measurements was calculated. Paw withdrawal latency was recorded for each animal before carrageenan administration and at 1, 2, and 3 h afterward [70].

#### 4.5.5. Oxaliplatin-Induced Neuropathy

Neuropathy was induced by administering a single dose (10 mg/kg) of Oxaliplatin (OXPT) dissolved in a 5% glucose solution (Polfa Kutno, Poland). To assess the response to mechanical stimuli, the von Frey test was carried with application of the electronic von Frey unit (Bioseb, France). The apparatus contained a single flexible filament, which was used to apply increasing force ranging from 0 to 10 g to the plantar surface of the mouse’s hind paw. The crossing of pain threshold led to the paw withdrawal. The mechanical pressure that evoked the nocifensive response was subsequently recorded. The measurement was performed before the OXPT administration and 7 days afterwards. The compounds were administrated to the animals with developed mechanical allodynia observed as a statistically significant decrease in pain threshold. On the day of the experiment, each mouse was placed in an observation chamber with a wire mesh bottom and was allowed to habituate for 1 h. After the habituation period, the mouse pain threshold was tested 3 times alternately with at least a 30 s gap between each measurement. The mean of these three consecutive measurements was taken as a baseline value. The mice with tactile allodynia were i.p. pretreated with the tested compound or their combinations, and 30 min later the animals were tested again according to the same procedure [71].

#### 4.5.6. Histamine-, Chloroquine- and SLIGR-L-Induced Pruritus

The area (around 2 cm^2^) of the nape of mice was shaved at least 2 days prior to experiments. On the day of the experiment, each mouse was individually placed in plastic chambers (15 × 15 × 30 cm). After the 30 min period of habituation, the mice received an intradermal (i.d.) injection of histamine dihydrochloride (10 µmol/site), chloroquine (CQ, 200 µg/site), or SLIGRL-NH2 (40 nmol/site), which were dissolved in physiological saline and administrated at the constant volume of 20 μL. Immediately following the injection, scratching (series of movements considered as a single scratching bout usually consisted of three single scratches) of the injected site by the hind paw was counted for 60 min, 30 min, or 45 min after histamine, CQ, and SLIGRL-NH2 injection, respectively [72,73,74].

## 5. Conclusions

This study provides evidence that E153 exerted both analgesic and antipruritic activities in animal models of pain and pruritus of different origins and mechanisms. The analgesic effect was dependent on H3R and sigma 1 receptor blockade, while the inhibition of the pruriceptive response was, at least in part, dependent on sigma 1 blockade.

Future research on E153 activity will focus on other molecular mechanisms, such as opioid kappa receptors, MAO-B, and BuChE inhibition. Another direction of activity of the compound, such as its influence on cognition, memory, and neurodegenerative disorders, will be tested. Moreover, the activity of the compound after oral administration will be evaluated, and methods of improving its pharmacokinetic properties, such as drug delivery systems using nanotechnology, will be developed.

## Figures and Tables

**Figure 1 pharmaceuticals-16-01481-f001:**
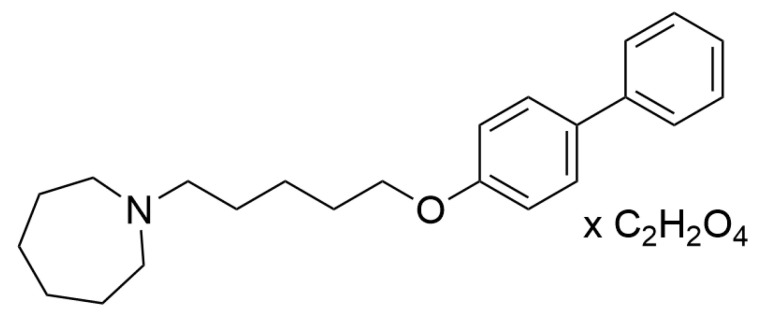
The chemical structure of **E153**.

**Figure 2 pharmaceuticals-16-01481-f002:**
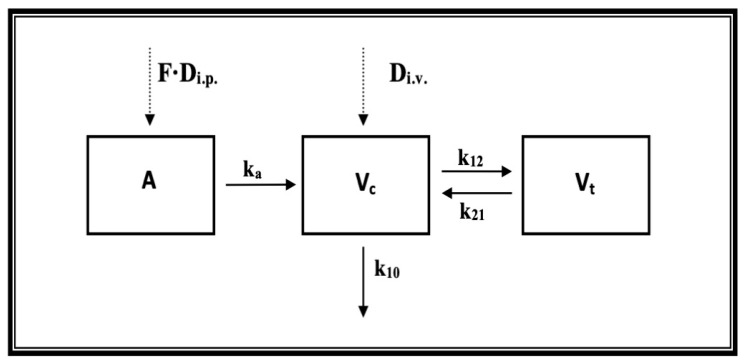
Pharmacokinetic model of E153 following i.v. and i.p. administration (F—fraction of a dose absorbed, A—amount of drug in the absorption compartment, D—dose, k_a—_first-order absorption rate constant, V_c_ and V_t_—volume of the central and tissue compartments, respectively, k_10_—first-order elimination rate constant, k_12_ and k_21_—first-order distribution rate constants).

**Figure 3 pharmaceuticals-16-01481-f003:**
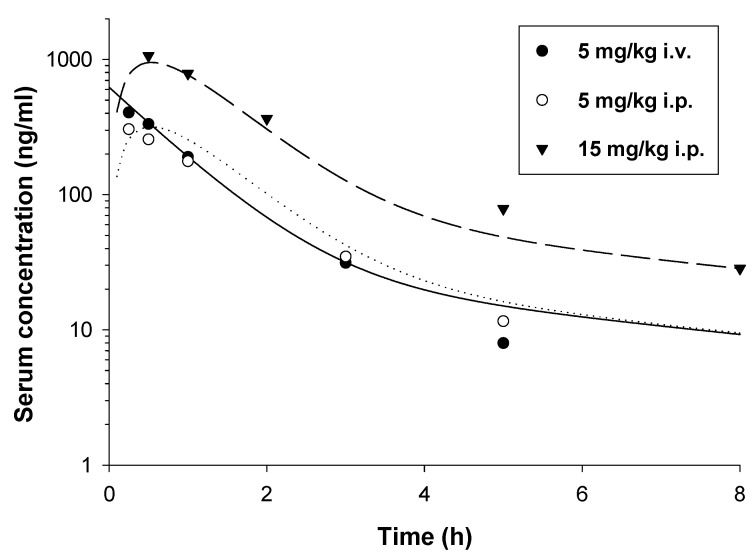
Observed (symbols) and pharmacokinetic model predicted (lines) concentration versus time profiles of E153 following i.p. and i.v. administration in mice.

**Figure 4 pharmaceuticals-16-01481-f004:**
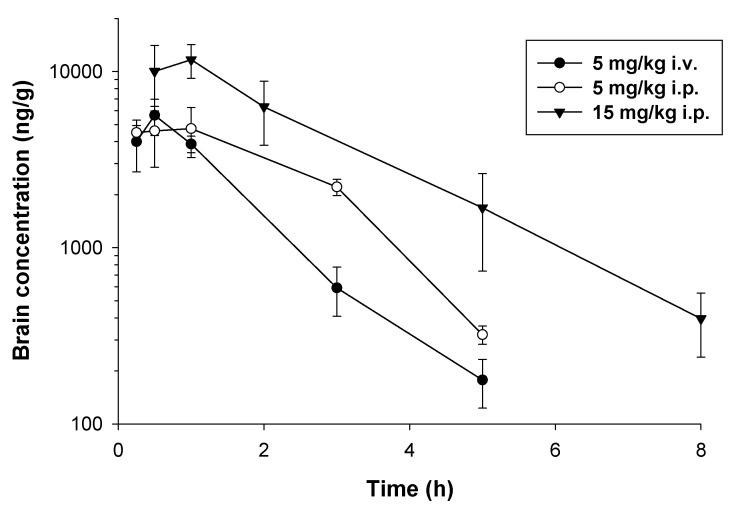
Brain concentration (the mean ± SD) versus time profiles of E153 following i.v. and i.p. administration to mice (*n* = 3–5 animals per time point).

**Figure 5 pharmaceuticals-16-01481-f005:**
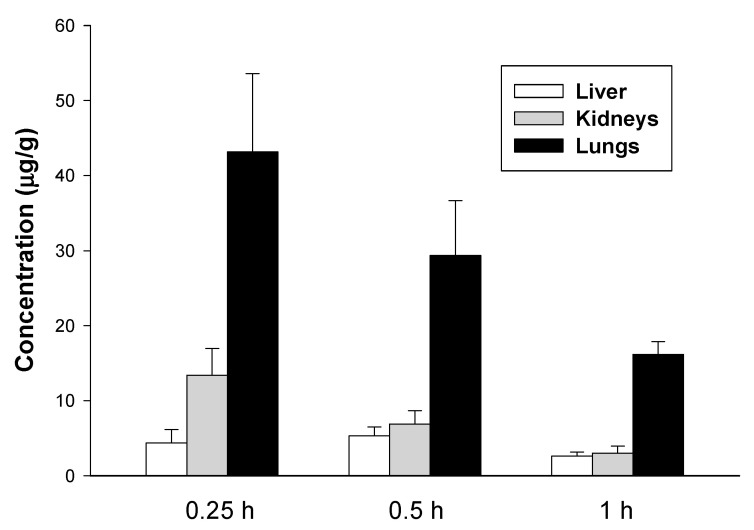
Tissue concentrations of E153 (the mean ± SD) at selected time points following i.v. administration of a dose of 5 mg/kg to mice (*n* = 3–5 animals per time point).

**Figure 6 pharmaceuticals-16-01481-f006:**
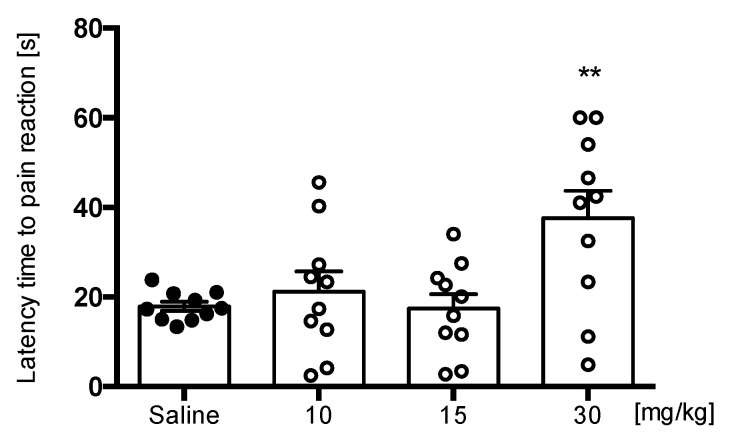
The effect of E153 on the latency time to pain reaction in the hot plate test (thermal stimulus). The test compound or saline were administered 30 min intraperitoneally (i.p.) before the test. The results are presented as bar plots showing the mean ± SEM. Statistical analysis: one-way ANOVA followed by Dunnett’s post hoc test, ** *p* < 0.01, *n* = 10 mice per group.

**Figure 7 pharmaceuticals-16-01481-f007:**
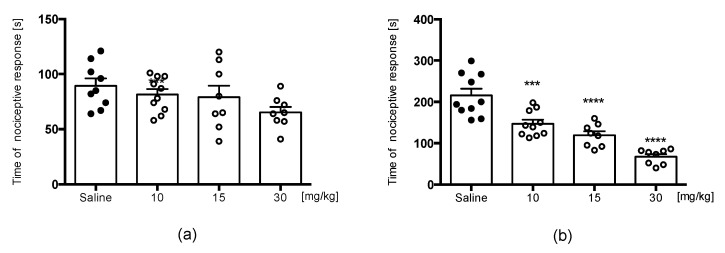
The impact of E153 on the duration of licking and biting behavior was assessed in both the acute phase (0–5 min following formalin injection, depicted in (**a**)) and the late phase (15–30 min after formalin injection, depicted in (**b**)). The test compound or a saline solution was administered intraperitoneally (i.p.) 30 min before the test. The results are illustrated in bar graphs, displaying the mean values along with the standard error of the mean (SEM). Statistical analysis was performed using a one-way ANOVA, followed by Dunnett’s post hoc test, where *** indicates *p* < 0.001, **** indicates *p* < 0.0001, and each group consisted of 8–10 mice.

**Figure 8 pharmaceuticals-16-01481-f008:**
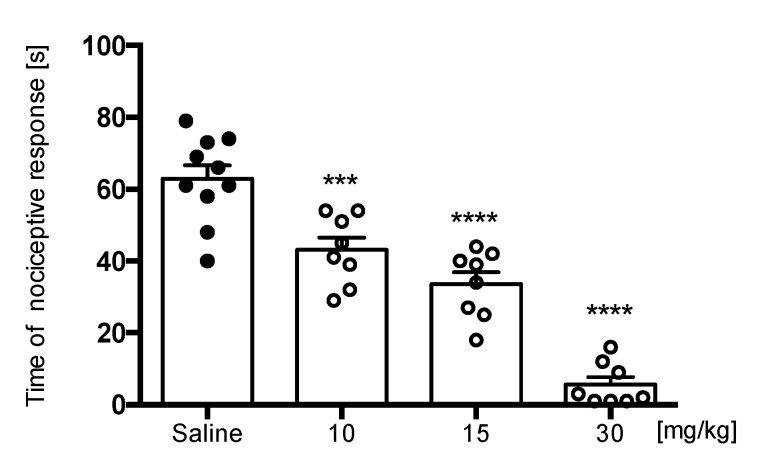
The effect of E153 on the duration of nociceptive response in capsaicin-induced pain. The test compound or saline were administered 30 min intraperitoneally (i.p.) before the capsaicin injection. The results are presented as bar plots showing the mean ± SEM. Statistical analysis: one-way ANOVA followed by Dunnett’s post hoc test, *** *p* < 0.001, **** *p* < 0.0001, *n* = 8–10 mice per group.

**Figure 9 pharmaceuticals-16-01481-f009:**
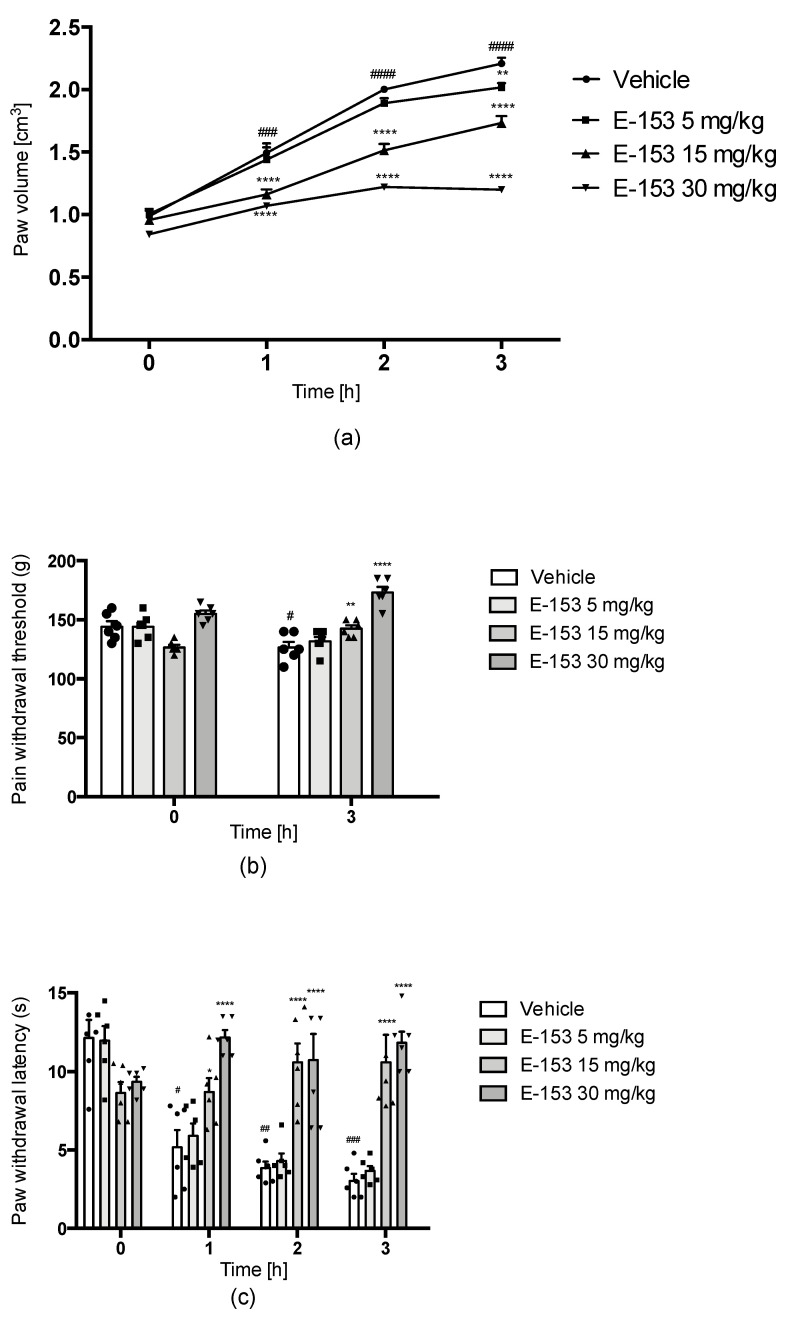
Influence of compound E153 on symptoms of carrageenan-induced inflammation such as edema (**a**), mechanical hyperalgesia (**b**), and on thermal hyperalgesia (**c**). Data are expressed as means ± S.E.M. for 6 animals. Time 0—the basic reaction considered as the pain threshold before carrageenan administration. Statistical significance compared to vehicle-treated animals: * *p*  <  0.05, ** *p* < 0.01, **** *p* < 0.0001. Statistical analysis: two-way ANOVA post hoc Bonferroni test. Statistical significance compared to initial value (0 time point) in vehicle-treated group: # *p* < 0.05, ## *p*  < 0.01, ### *p* < 0.001 #### *p* <  0.0001. Statistical analysis: one-way ANOVA followed by Dunnett’s post hoc test (**a**,**c**) or *t*-test (**b**).

**Figure 10 pharmaceuticals-16-01481-f010:**
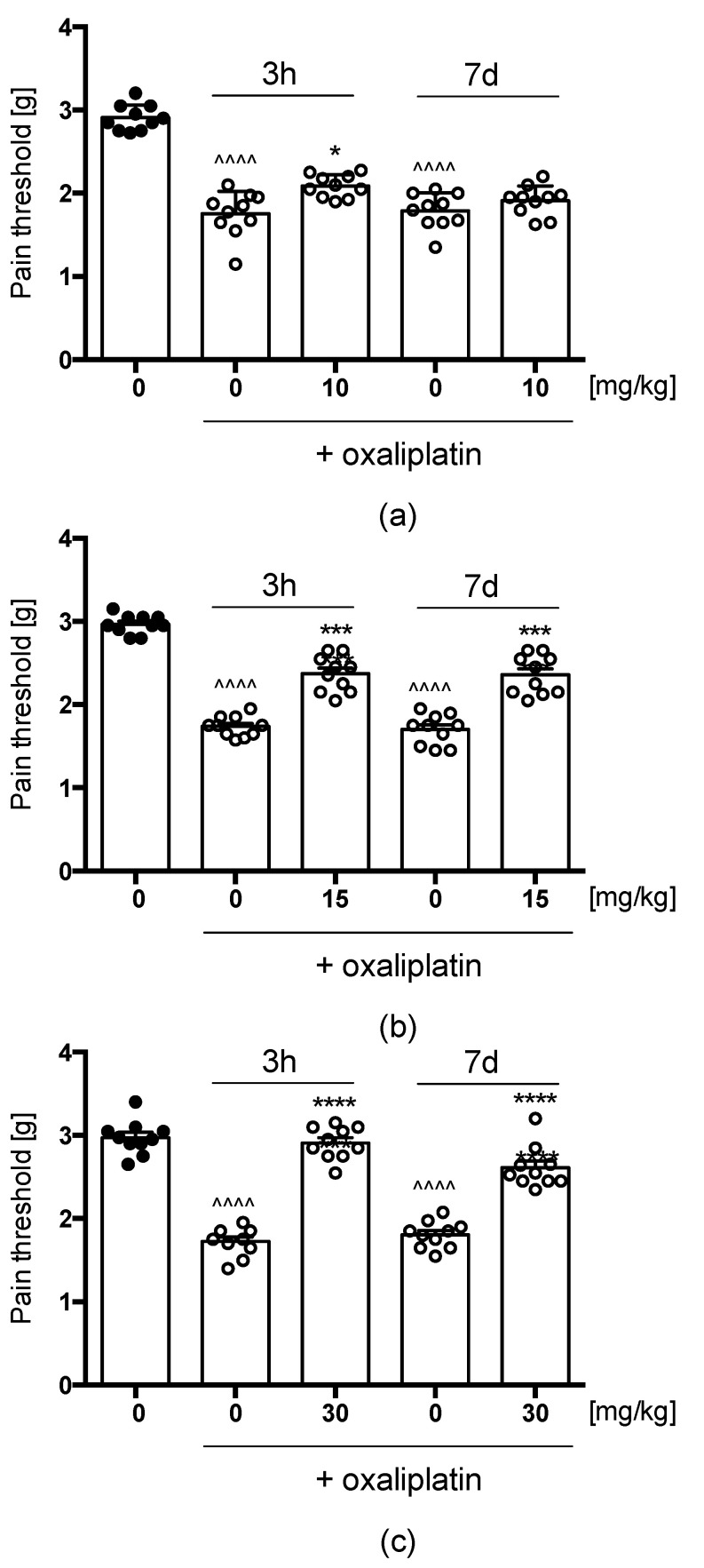
Antiallodynic effects of E153 in the tactile allodynia in OXPT-induced peripheral neuropathy. The compound was administered at the doses of 10 (Panel (**a**)), 15 (Panel (**b**)), and 30 mg/kg (Panel (**c**)) 30 min before the evaluation in the von Frey test carried out 3 h (3 h) and 7 days (7 d) after oxaliplatin injection. Statistical analysis included repeated measures analysis of variance (ANOVA), followed by Dunnett’s post hoc comparison: * *p*  <  0.05, *** *p* < 0.001, **** *p*  <  0.0001 when results compared to oxaliplatin-treated group (0 mg/kg + oxaliplatin), and ^^^^ *p*  <  0.0001 when results compared to vehicle-treated (1% Tween 80) mice (0 mg/kg), *n* = 10 mice per group.

**Figure 11 pharmaceuticals-16-01481-f011:**
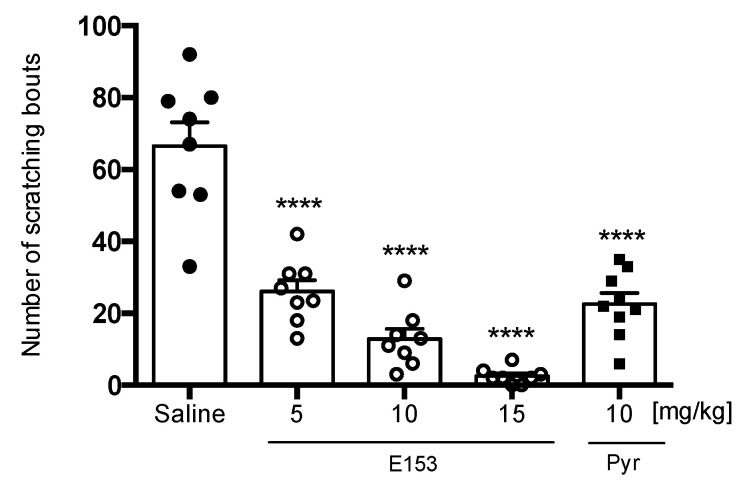
The effect of E153 and pyrilamine (Pyr) on the amount of scratching bouts during observation (60 s) in histamine-induced pruritus. The test compound or saline were administered 30 min intraperitoneally (i.p.) before the intradermal (i.d.) injection of histamine. The results are presented as bar plots showing the mean ± SEM. Statistical analysis: one-way ANOVA followed by Dunnett’s post hoc test, **** *p* < 0.0001, *n* = 8 mice per group.

**Figure 12 pharmaceuticals-16-01481-f012:**
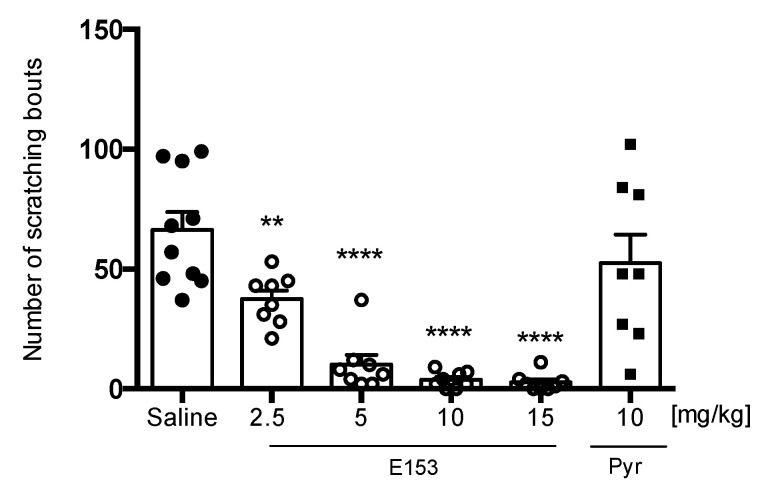
The effect of E153 and pyrilamine (Pyr) on the number of scratching bouts during 30 min long observation in CQ-induced pruritus. The test compound or saline were administered 30 min intraperitoneally (i.p.) before the intradermal (i.d.) injection of CQ. The results are presented as bar plots showing the mean ± SEM. Statistical analysis: one-way ANOVA followed by Dunnett’s post hoc test, ** *p* < 0.01 **** *p* < 0.0001, *n* = 8–10 mice per group.

**Figure 13 pharmaceuticals-16-01481-f013:**
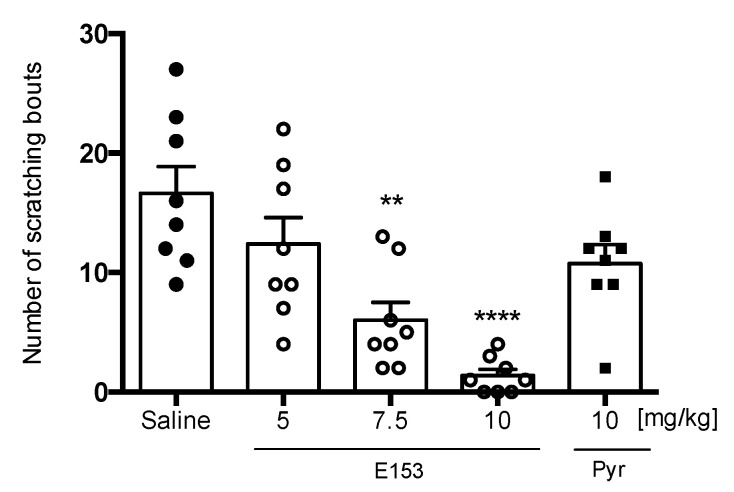
The effect of E153 and pyrilamine (Pyr) on the number of scratching bouts during 45 min long observation in SLIGR-L-induced pruritus. The test compound or saline were administered 30 min intraperitoneally (i.p.) before the intradermal (i.d.) injection of SLIGR-L. The results are presented as bar plots showing the mean ± SEM. Statistical analysis: one-way ANOVA followed by Dunnett’s post hoc test, ** *p* < 0.01 **** *p* < 0.0001, *n* = 8–10 mice per group.

**Figure 14 pharmaceuticals-16-01481-f014:**
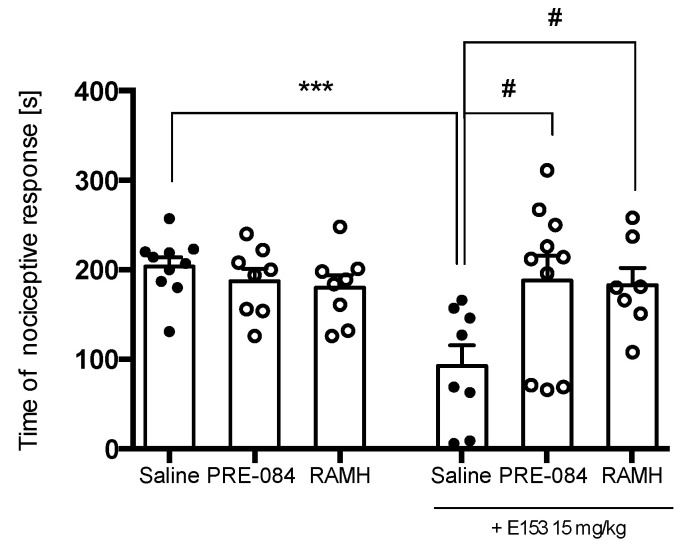
The influence of PRE-084 (15 mg/kg), the sigma 1 agonist, and (R)-(-)-α-methylhistamine (RAMH) (15 mg/kg), histamine H_3_R agonist, on the analgesic effect of E153 in the late phase of the formalin test. The test compound or saline were administered 30 min intraperitoneally (i.p.) before the test. PRE-084 was administered subcutaneously 10 min before the administration of E153. RAMH was administered intraperitoneally 10 min before the administration of E153. The results are presented as bar plots showing the mean ± SEM. Statistical analysis: one-way ANOVA followed by Dunnett’s post hoc test, *** *p* < 0.001 when compared to Saline treated animals, # *p* < 0.1 when compared to E153 (15 mg/kg) treated animals, *n* = 8–10 mice per group.

**Figure 15 pharmaceuticals-16-01481-f015:**
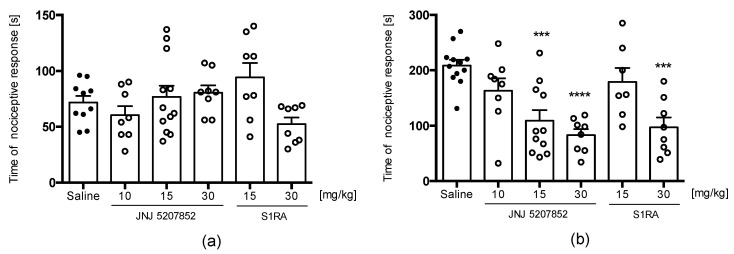
The effect of JNJ 5207852 and S1RA on the duration of licking/biting behavior in acute phase (0–5 min after formalin injection, (**a**)) and in the late phase (15–30 min after formalin injection, (**b**)). The test compound or saline were administered 30 min intraperitoneally (i.p.) before the test. The results are presented as bar plots showing the mean ± SEM. Statistical analysis: one-way ANOVA followed by Dunnett’s post hoc test, *** *p* < 0.001, **** *p* < 0.0001, *n* = 7–11 mice per group.

**Figure 16 pharmaceuticals-16-01481-f016:**
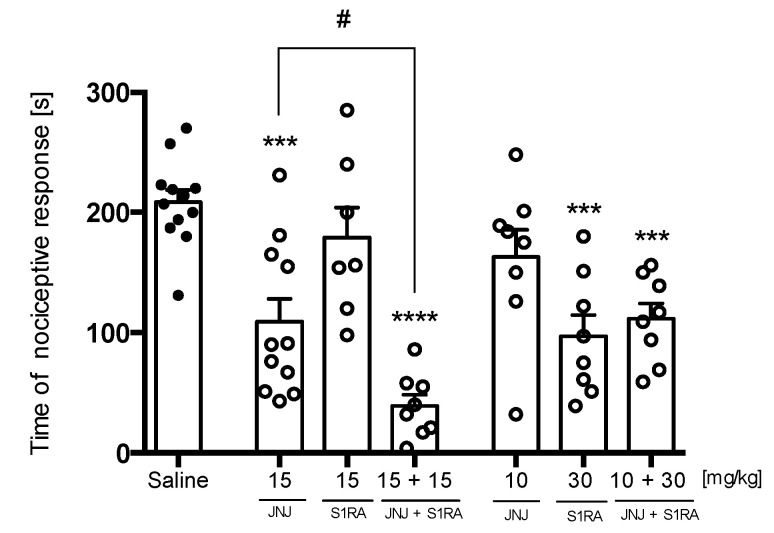
Mutual effects of a selective H_3_R antagonist (JNJ 5207852) and a sigma 1 antagonist (S1RA) on analgesic activity in the late phase of the formalin test. The test compounds or saline were administered 30 min intraperitoneally (i.p.) before the test. The results are presented as bar plots showing the mean ± SEM. Statistical analysis included one-way ANOVA followed by Dunnett’s post hoc test: *** *p* < 0.001, **** *p* < 0.0001 when compared to saline-treated animals; # *p* < 0.1 when compared to JNJ 5207852 (15 mg/kg)-treated animals to JNJ 5207852 (15 mg/kg) and S1RA (15 mg/kg)-treated animals, *n* = 8–12 mice per group.

**Figure 17 pharmaceuticals-16-01481-f017:**
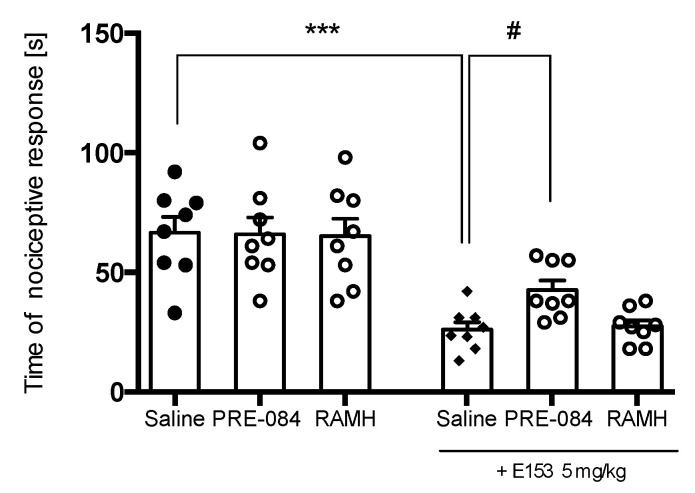
The influence of PRE-084 (15 mg/kg), the sigma1 agonist and (R)-(-)-α-methylhistamine (RAMH) (15 mg/kg), histamine H_3_R agonist on the antipruritic effect of E153 in the histamine-induced pruritus. The test compound or saline were administered 30 min intraperitoneally (i.p.) before the test. PRE-084 was administered subcutaneously 10 min before the administration of E153. RAMH was administered intraperitoneally 10 min before the administration of E153. The results are presented as bar plots showing the mean ± SEM. Statistical analysis included one-way ANOVA followed by Dunnett’s post hoc test: *** *p* < 0.001 when compared to Saline treated animals, # *p* < 0.1 when compared to E153 (5 mg/kg) treated animals, *n* = 8–10 mice per group.

**Table 1 pharmaceuticals-16-01481-t001:** Results of in vitro binding assays of E153. Screening at the concentration of 1 μM.

Human Histamine H_3_ Receptor ^a^	Human Sigma Receptors		Human Opioid Receptors		
	Sigma 1 ^b^	Sigma 2 ^c^	Kappa ^d^	Delta ^e^	μ ^f^
K_i_ = 33.9 ± 8.1 nM ^g^	IC_50_ = 2.4 nM (K_i_ = 1.2 nM)	(101.7%) ^h^	(52.1%) ^h^	(−5.3%) ^h^	(11.2%) ^h^

^a^ [^3^H]N^α^-methylhistamine (agonist) in HEK293 cells; ^b^ [^3^H](+)pentazocine (agonist) in Jurkat cells; ^c^ [^3^H]-1,3-di-o-tolylguanidine ([^3^H]DTG) (agonist) in the presence of unlabeled (+)-pentazocine in Jurkat cells; ^d^ [^3^H]U69593 (agonist) in RBL cells; ^e^ [^3^H]DADLE (agonist) in RBL cells; ^f^ [^3^H]DAMGO (agonist) in HEK293 cells; ^g^ data from [23]; ^h^ % of inhibition at 1 μM; mean values of two independent experiments.

**Table 2 pharmaceuticals-16-01481-t002:** Pharmacokinetic parameters of E153 assessed for i.v. and i.p. administration simultaneously using a two-compartment model.

Parameter (Unit)	Estimate	CV (%)
V_c_ (L/kg)	6.39	19.30
k_10_ (h^−1^)	1.00	23.61
k_01_ (h^−1^)	2.32	28.14
k_12_ (h^−1^)	0.33	28.66
k_21_ (h^−1^)	0.19	72.20
F	0.90	11.55

CV—coefficient of variation.

## Data Availability

Data are contained within the article and Appendix A.

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
