# Peer review of "Efficacy of the Multi-Target Compound E153 in Relieving Pain and Pruritus of Different Origins"

_pharmaceuticals, 2023, doi:10.3390/ph16101481_

Round 1

Reviewer 1 Report

Dear Dr.,

Title: Efficacy of the multi-target compound E153 in relieving pain and pruritus of different origin

Manuscript ID: pharmaceuticals-2591879

Overall comments: Mogilski et al. described in this manuscript: the role of E153 in the amelioration of pain and pruritus. Further, the author described the pharmacological tools of PRE-084, RAMH, JNJ 5207852, and S1RA for the determination of histamine H3 and sigma 1 receptor's role in the analgesic and antipruritic effects. The major limitation of clear description is missing in the introduction and discussion sections. The overall manuscript is good and it can help those working in this field of research.

Specific comments:

1.      The introduction and discussion sections are too lengthy; and need to be written concise manner with relevant information to the research concept.

2.      Result section headings and subheadings must be rewritten as per the guidelines.

3.      Section 2.3.3.; must be made more clear statements.

4.      In methodology a number of animals and groups were used for Pharmacokinetic analysis, In Vivo Pharmacological Studies need to be described clear manner.

5.      The limitation must be described in the discussion section.

6.      Future directions must be mentioned in the conclusion section.

7.      References are too old, need to cite the recent & relevant references in the text as well as in the list as per the text information.

Minor comments

1.      The text information needs to be rearranged logical manner as per the author's instructions.

2.      Text alignment and typo errors need to be rectified.

3.      References are not correct format in the text. Need to rectify this error.

*****

Minor English correction is required.

Reviewer 2 Report

The paper of Dr.  Mogilski is an interesting research on effects of a new potential analgesic and anti-pruritic agent, E153. The study is done in vivo in rats and mice; the researchers aimed to assess the quantitative effects on itching and/or nociceptive responses, modified by E153 alone, or in combination with H3 histamine antagonist or Sigma 1 antagonists.

The paper is well designed, and presents a large dataset. It is interesting for a wide auditorium of pre-clinical scientists.  

A major point to improve the manuscript would be an addition of experimental or literature data on the central effects of E153. Since the authors assess mainly the behavioural responses of test animals, it is a need to understand a level of vigilance, modified (or not) by the agent.

Minor points are the following:

line 167 – should the substance be mentioned as a mixed H3/H4 antagonist?

line 170 – it is not clear, was the hepatotoxicity found for this substance? or was it low? please, add a word to specify

Figure 2 – not all abbreviations used are deciphered in the Legend (please, add D’s)

-        what are the metabolites of E-153?

-        Figure 4,5: what does mean the number of mice (n=3-5) in the Legend? is it a number of mice per point? please, specify the method applied – is it a wet tissue concentration, or what?

-        If E 153 accumulates mainly in lungs (Figure 5), then does it affect respiration?

-        Lines 288-289 :” The compound significantly  decreased the behaviour of paw licking or biting in that test at all doses administrated (10, 289 15 and 30 mg/kg) [F(3,30) = 53.20, p<0.0001]. (Figure 8.)” – please, provide or refer to the data, describing an absence of motor dis-coordination, induced by E153 administration

-        Which behavioral elements were registered by blind observers? sample video or drawn schematic would be very useful

Reviewer 3 Report

I read with great interest this article that presents globally a solid design and well displayed results. Conclusions are supported.

I have only minor revisions to suggest:

Please comment on the pruritus in PsO patients that cleared after treatment [10.1080/09546634.2020.1840502], in cancer treated with chemotherapies [10.1371/journal.pone.0255716]

Please ope the possibility in conclusion to use nanotechnologies as carriers.

Round 2

Reviewer 2 Report

the manuscript has been improved and can be published in the present form